# A Bi-Objective Home Health Care Routing and Scheduling Problem under Uncertainty

**DOI:** 10.3390/ijerph21030377

**Published:** 2024-03-21

**Authors:** Jiao Zhao, Tao Wang, Thibaud Monteiro

**Affiliations:** 1INSA Lyon, Université Lumière Lyon 2, Université Claude Bernard Lyon 1, Université Jean Monnet Saint-Etienne, DISP UR4570, 69621 Villeurbanne, France; jiao.zhao@insa-lyon.fr; 2Université Jean Monnet Saint-Etienne, INSA Lyon, Université Lumière Lyon 2, Université Claude Bernard Lyon 1, DISP UR4570, 42300 Roanne, France; tao.wang@univ-st-etienne.fr

**Keywords:** home health care, routing and scheduling problems, bi-objective optimization, adaptive large neighborhood search, uncertainty

## Abstract

Home health care companies provide health care services to patients in their homes. Due to increasing demand, the provision of home health care services requires effective management of operational costs while satisfying both patients and caregivers. In practice, uncertain service times might lead to considerable delays that adversely affect service quality. To this end, this paper proposes a new bi-objective optimization problem to model the routing and scheduling problems under uncertainty in home health care, considering the qualification and workload of caregivers. A mixed-integer linear programming formulation is developed. Motivated by the challenge of computational time, we propose the Adaptive Large Neighborhood Search embedded in an Enhanced Multi-Directional Local Search framework (ALNS-EMDLS). A stochastic ALNS-EMDLS is introduced to handle uncertain service times for patients. Three kinds of metrics for evaluating the Pareto fronts highlight the efficiency of our proposed method. The sensitivity analysis validates the robustness of the proposed model and method. Finally, we apply the method to a real-life case and provide managerial recommendations.

## 1. Introduction

The Home Health Care (HHC) industry has been fast-growing worldwide in recent years with the development of information technology and transportation systems. HHC companies provide services at patients’ homes, ranging from nursing care to specialized medical services [1]. Patients can thus receive treatment in familiar surroundings. This also helps to decrease hospital admissions and duration of hospital stays [2]. Demand for HHC has been increasing in recent years as the elderly increasingly prefer to grow old in their own home, underscoring the critical need for the efficient organization of caregivers’ daily activities.

The integration and coordination of this health service logistic network is a complex task, and managers have to face many logistics decisions. Network design, transportation management, staff management, and inventory management are all described as the set of decision problems related to design and operation in HHC [3]. Depending on the time horizon from long term to short term, the planning horizon can be categorized into three levels: the strategic level, the tactical level, and the operational level [4]. The strategic-level decisions involve defining facility locations, patient districts, transportation modes, staffing, and service levels. The fleet assignment to patients’ districts, the shift scheduling, and the definition of inventory policies are considered at the tactical level. The decisions at the operational level include staff assignment and routing, as well as inventory control. A fourth level has recently been recognized as a real-time level, where very short-term planning is needed, according to actual execution.

The short-term daily activities in HHC services are shown in Figure 1. Once patients receive their medical treatment prescriptions, service times and staff qualifications are defined, and managers assign qualified caregivers to patients. They draw up the caregivers’ schedule based on the services requested and on the patients’ available times. Finally, each caregiver delivers their medical service, starting from the home health care facility and visiting patients in a planned sequence before returning to the facility.

These daily activities in HHC refer to transportation management and can be planned at the tactical and operational levels. This planning can be modeled as a Home Health Care Routing and Scheduling Problem (HHCRSP). Caregivers should arrive and leave within the time frame of patients’ availability, corresponding to a fixed time interval for receiving the care service, called a time window. A hard time window ensures that the patients are visited within their time windows. Caregivers are not permitted to perform the service if they arrive or leave outside of these time windows. A soft time window can be violated at the cost of a penalty.

When determining routes and schedules, decision-makers strive to identify the most cost-effective routes while taking into account time windows. This is crucial as transportation services constitute a substantial portion of a company’s expenses and can impact operational efficiency. However, both patient satisfaction and caregiver satisfaction also play an important role in HHC companies’ competitiveness. HHC companies may not be capable of providing service for all patients in their available time frames. All patients can, however, receive care if some of them accept visits outside of their available time frames. The less that arrival and departure times are outside the time window, the more satisfied patients and caregivers will be. It is therefore particularly important to strike a balance between costs and quality, which has motivated us to propose two objectives.

In practical care services, caregivers must possess appropriate qualifications to provide medical care that aligns with the patient’s conditions. The level of treatment required should correspond to the capabilities of the caregivers. Fairness of caregivers’ workload also has to be ensured to avoid overwork or refusal of certain treatment tasks. These commonly considered features, including different staff qualifications and workload balance for caregivers, contribute to the service quality but make the model more complex.

In addition to the operational constraints, uncertainty is prevalent in such applications. A caregiver may arrive or leave a patient’s home earlier or later than the previous appointment time as the schedule and the route are static. The time that a caregiver spends on the road changes little, except for instances of major road accidents. The arrival time and service end time are hard to predetermine mainly because the service time of each patient is related to their physical condition, which is often unstable, especially when emergencies or acute diseases are involved. Moreover, providing services to address social isolation among elderly patients adds to the unpredictability of service time. Individual needs and preferences for social interaction and companionship can vary significantly. Some patients may require more time and attention for meaningful conversations or activities than the planned service times. If caregivers arrive prematurely due to the preceding service ending sooner, they have to wait. This inadvertently extends their working hours. If the actual service time is longer than the planned service time, the caregiver may arrive too late at the next patient’s home. This case where the patient cannot receive service on time degrades the service quality. In diabetes management, maintaining strict adherence to time schedules is crucial for stable blood glucose level control [5]. Similarly, time-regulated nutritional support is needed in palliative care [6]. It is therefore essential to plan robust routes and schedules for real-life home health care activities.

To conclude, there is an urgent need to define effective routes and schedules while enhancing the quality of service under uncertain service times. To address this, a new bi-objective Mixed-Integer Linear Programming (MILP) model is developed, integrating routing and appointment scheduling. Due to the complexity of solving this model for practical-sized instances, the Adaptive Large Neighborhood Search within Enhanced Multi-directional Local Search (ALNS-EMDLS) has been developed and is proposed to speed up problem-solving and provide the approximation Pareto front. Additionally, a stochastic version of ALNS-EMDLS is introduced to generate robust solutions in the face of uncertain service times. These solutions may not be optimal for every service time scenario, but they perform better on average than the deterministic method.

The rest of this paper is organized as follows. Section 2 reviews the literature related to vehicle routing and scheduling problems in home health care activities. Section 3 addresses the model with two objective functions. A scenario-based multi-objective solution methodology is proposed in Section 4. The parameters setting, computational experiments, and results analysis are presented in Section 5. In Section 6, the parameters inherited from the sensitivity analysis are utilized in a real-world application, which results in managerial suggestions. The conclusion of this paper and the perspective of future studies are summarized in the Section 7.

## 2. Related Works

In this paper, we focus on the HHCRSP within a daily planning horizon, which is a variant of the Vehicle Routing Problem with Time Windows (VRPTW) [7]. The VRPTW has many applications such as telecommunication, waste collection, and cross-docking [8,9]. Determining the optimal solution to VRPTW is NP-hard. Solving the HHCRSP consists of designing optimal delivery routes from a central location to a set of geographically distributed patients with various constraints. This entails optimization problems that are complex and therefore of particular interest to stakeholders. It differs from VRPTW because of the features [10]: (1) the temporal dependency and the disjunctive nature of services; (2) the continuity, given that patients are assigned to a restricted set of caregivers; and (3) caregivers’ skills and patients’ requests. Various constraints and objectives are used in different studies due to the home care policies of the country under study. Exact, heuristic, and approximate methods have been proposed to solve related problems [11].

### 2.1. Operational Constraints

Liu et al. [12] addressed a routing and scheduling problem for home care workers with the consideration of lunch break requirements. They found an optimal solution by the Branch and Price (B&P) method. Shahnejat-Bushehri et al. [13] took into account Time Window (TW), Quality of Caregivers (QC), precedence, and synchronization constraints. A parameter was used to indicate whether a caregiver had the necessary qualifications to provide a certain service. Simulated Annealing (SA) and Tabu Search (TS) were applied in two phases. A Variable Neighborhood Search (VNS)-based heuristic was used to obtain a feasible solution that had to observe assignment constraints, working time restrictions, time windows, and mandatory break times [14]. Constraint Programming (CP) and TS were used to solve the routing problem while considering the problem of matching the skills of caregivers and the patient’s conditions [15]. Workload Balance (WB) was considered as the fair workload assignment to each caregiver in [16]. The constraints in existing research also include visits incompatibilities [17], multiple modes of transportation [18], and time-dependent travel times [19].

In most other studies, the time window is used to limit the service starting time. The service starting time outside the time window leads to a penalty due to patients’ dissatisfaction. However, it is more reasonable that the time window is defined as the time frame during which the patient is available. There will also be a penalty if caregivers perform a service and then leave the patients’ homes outside the time windows. We, therefore, define patients’ and caregivers’ satisfaction as minimizing the segmented penalty due to arrival times and departure times being out of the time windows. If the penalty is continuous, only one patient is likely to endure a long delay. The segmented penalty promotes a more equitable distribution of service punctuality and increases the fairness in scheduling for patients and caregivers. Different levels of caregivers are required by patients depending on the severity of their conditions. We assume a fixed number of caregivers to be assigned to the daily schedule. The number of patients to be served by one caregiver is limited for the sake of balancing the workload. We first consider these properties together.

### 2.2. Multi-Objective Optimization

Decision-makers balance various factors, including cost, requiring the simultaneous optimization of multiple objectives, an approach facilitated by Multi-Objective Optimization (MOO) [20]. There is seldom a single global solution, and it is often necessary to determine a set of points that all fit a predetermined definition of an optimum (for more concepts, see [21]). One of the most intuitive methods is to optimize the weighted sum of all the objective functions as a single objective optimization. It is implemented simply but is highly dependent on weights. The bounded objective function method minimizes the single most important objective function, while others are used to form additional constraints. The bounded value and the most important objective function are hard to pre-select. The optimization process of the lexicographic method is carried out individually on each objective function following the order of importance and stops when a unique solution is obtained. In a game theoretic approach, objective functions are assumed to be the players that are ultimately controlled by the decision-maker and can be expected to reach an agreement, meaning the game is cooperative. This was proposed in [22], and an improvement was stated in [23]. A single solution method can be used if the decision-makers’ preference is known. However, if they cannot explicitly express their preference, it is preferable to provide them with a range of solutions to choose from. Some multi-objective optimization algorithms are capable of obtaining Pareto solutions, which represent the set of non-dominated solutions in a multi-objective optimization problem. These solutions are essential for decision-makers to evaluate trade-offs between conflicting objectives. A normal-boundary intersection is a technique intended to find the portion that contains the Pareto optimal solutions, delineating the boundary of the set of attainable objective vectors [24]. A genetic algorithm is also suitable to obtain Pareto fronts since it can process a set of solutions in parallel. Non-dominated sorting genetic algorithms and other algorithms based on the genetic algorithm combine the use the random numbers and information from previous iterations to evaluate and improve a population of points [25]. Recently, the Pareto Q-learning algorithm was used for solving MOO, learning deterministic, non-stationary, and non-dominated multi-objective policies while mapping the entire Pareto front [26,27].

Recent advances in HHC research have moved to explore the multi-objective optimization method to obtain the Pareto front instead of using a weighted objective function. Decerle et al. [28] combined distance and visit penalties into a single objective function, finding genetic algorithms and local searches that yielded instance-flexible results. In [29], the weighted sum of travel time, a score of continuity of care, overtime, idle time, and penalty for unscheduled patients, was minimized. Yang et al. [30] minimized travel costs, inconsistency, and workload balance using an artificial bee colony metaheuristic to generate non-dominated solutions. Braekers et al. [31] analyzed the trade-off between operating costs and service levels. A bi-objective optimization model was developed to address the travel costs and downgrading costs of nurses in [32]. This model was solved by an ϵ-constraint-based approach. Fathollahi-Fard et al. [33] considered travel costs and CO_2_ emissions as objective functions, using hybrid versions of metaheuristics and developing four fast heuristics for Pareto optimal solutions.

Although improving service quality and patient satisfaction is as important as reducing costs, there is still less related research on the subject. This has motivated us to introduce a bi-objective model to reconcile the interests of different stakeholders in HHC. We do not depend on the preference of decision-makers (for example, weights assigned to the objectives) to aggregate the objectives into one. Decision-makers can select their preferred solution from a Pareto optimal set according to different operational situations.

Table 1 summarizes the research that built deterministic models for routing problems in HHC. Compared with the deterministic models in recent research, our work takes into account the time windows, workload balance, and skills matching. We propose a novel approach ALNS-EMDLS. It is easy to implement and ignores the gradient information and the nature of objective functions and constraints. It contains fewer hyperparameters compared to genetic algorithms.

### 2.3. Uncertainties

Some parameters that are represented in a stochastic manner can model uncertainties such as service times, travel times, and demands of patients. Two common models used in general formulations are the Chance Constrained Programming (CCP) model and the Stochastic Programming with Recourse (SPR). Li et al. [34] conducted a comparative study of these two models for the vehicle routing problem under uncertain travel times and service times. Based on the results obtained by TS, the authors concluded the CCP might not be a suitable model for the target problem. This is attributed to the computational challenges arising from the stringent constraints imposed by the confidence levels. Consequently, the stochastic programming model has been chosen for handling the uncertain service times in our study.

The robust optimization aims to obtain robust solutions that remain relatively unchanged under uncertainties. The uncertainties can be modeled deterministically, probabilistically, or possibilistically [35]. The uncertainties in HHC can be mainly quantified based on certain distributions [30,36,37], the triangular fuzzy numbers [38,39], the budget uncertainty polytopes [40], and a set of scenarios with fixed probabilities [41,42]. The worst-case philosophy and expected performance of all scenarios can be employed to construct the robust objective function of the original formulation. The former, though conservative, can yield impractical solutions when overly large domains are chosen. Therefore, we define the robust counterpart of the original objective function based on the expectation. We assume that the uncertainty in our study follows the normal distribution. If solutions are obtained with optimal expected values while involving uncertainties that are random variables or follow probability distributions, it can be also called stochastic optimization.

To solve the HHCRSP under uncertainties, some studies in the HHCRSP utilize numerical techniques to transform the robust optimization problem into a normal optimization problem by using strong mathematical assumptions. Yuan et al. [43] used an approximate formula to replace the expected penalty for late arrival and thus to reduce computing effort. In [30], the objective functions and constraints related to uncertain service time and travel time were considered and transformed into their deterministic equivalents based on uncertainty theory. This consistent HHCRSP was solved by an improved multi-objective Artificial Bee Colony (ABC) metaheuristic. The objective function was rewritten as a recursive function based on the theory of budget in [40]. However, this approach may be limited by the need for strong mathematical assumptions like first- or second-order derivatives, which may not always be accessible. In most studies, the robust objective function is typically calculated by the simulation method, and then exact, matheuristic, or metaheuristic methods are applied [36,37].

In other fields, some researchers apply the scenario-based method to solve a stochastic multi-objective model [44,45,46]. A multi-objective optimization problem under uncertainty in transmission expansion planning was proposed in [47]. The objective functions were the total cost, the robustness, and the flexibility criterion. The proposed process for solving this problem considered the performance of solutions in all scenarios simultaneously. It could be applied to a situation where there were not too many scenarios because the model needs to be optimized under each scenario. In the micro-grid operation field, Niknam et al. [48] modeled load demand, available output power, and market price, by means of scenario-based stochastic programming.

In our study, we develop the stochastic model and method to obtain robust routes and schedules. Given a certain distribution of the service time, we aim to optimize the expectation of the objective functions instead of using only the mean of the service times for a deterministic model. It is hard to calculate the integration of complex objective functions. Sampling from the distribution into a finite set also generates a big increase in the number of variables and constraints. Therefore, we propose a stochastic ALNS-EMDLS that combines a scenario-based method to deal with the uncertain service times.

Notably, we conclude the main research gaps by comparing our study with the following literature that is closely related. The authors of [43] modified the objective function and constraints with stochastic service times by an approximate formula, while the authors of [37] used Monte Carlo simulation. However, they only focused on single-objective optimization problems using Monte Carlo simulation or the scenario-based method. Our study is the first to apply the scenario-based stochastic method in addressing the multi-objective optimization problem in this domain. The scenario-based stochastic method is data-driven and enhances decision-making under uncertainty by evaluating multiple scenarios, allowing for more informed strategies [49]. The multi-objective optimization model and algorithm that we propose offers a broader range of choices where manager preferences are unknown.

The main contributions of our work are as follows:We develop a new bi-objective MILP model that aims to optimize the travel cost as well as the satisfaction of caregivers and patients, considering the alignment of caregivers’ qualifications with patients’ requirements as well as workload balance.We propose the ALNS-EMDLS to solve the problem. The effectiveness of the new approach is validated by experimental results thanks to the comparison with the Gurobi solver [50]. The stochastic ALNS-EMDLS is proposed to deal with uncertainties. The contrast between the stochastic and original versions demonstrates the stochastic method’s robustness.In order to refine and enhance the application of our method, we conduct a sensitivity analysis to identify suitable parameters and apply them to real-world data in a case study, providing actionable management recommendations to choose the suitable schedules.

## 3. Problem Statement

In this section, a mathematical model and its components representing the extension of VRPTW in HHC have been formulated. From the perspective of graph theory: let G=(V,A), where V={0}∪N={1,2…,n} is the vertex set and represents the depot and the patients. A={(i,j)∈V,i≠j} is the set of arcs. We aim to find minimum-cost routes that serve the vertices once and satisfy the side constraints of arc R⊆A.

It is assumed that all caregivers leave the depot at time 0. Caregiver k∈K starts from the facility, moves once to each patient, and returns to the depot. Each caregiver has the qualification level $Q_{k}$. Fixed levels of qualification Q={1,...,q} are defined in this paper. A visit is therefore allowed only if the patient’s requirement is lower or equal to the qualification level of the caregiver. The skill level of nurses is mainly based on their experience in dealing with complex cases. Patients are assured of receiving appropriate levels of care. The qualification matching ensures the service quality and safety.

Patients are spread across different locations. Each patient *i* has requirement $RC_{i}$ for a specific level of the caregiver. The patients’ requirement set is aligned with the qualification set of the caregivers. The patients indicate their preferred time for home care during the registration process. The time window $\left[e_{i},l_{i}\right]$ of the patient *i* is defined as the earliest time of starting service and the latest time of ending service that can be tolerated by the patient. If the caregiver arrives earlier than $e_{i}$, the service will start before reaching $e_{i}$; if the arrival is after $e_{i}$, the service begins immediately. The caregiver will leave immediately after serving for service time $\delta_{i}$.

The departure time of node *i* can be calculated as (Equation 1):(1)$$d_{ik}=\max\left(a_{ik},e_{i}\right)+\delta_{i}$$ The arrival time of patient *j* is calculated as (Equation 2):(2)$$a_{jk}=x_{ijk}\left(d_{ik}+t_{ij}\right).$$ Note that $a_{jk}$ is meaningless when patient *i* is not visited by caregiver *k*, that is, $x_{ijk}=0$. Caregivers who arrive too early may face long waiting times, while arriving too late can lead to decreased patient satisfaction. A penalty cost is introduced in objective functions when caregivers arrive or leave outside the time window. Minimizing the penalty can enhance service punctuality and reduce waiting times, thereby improving the satisfaction of both patients and caregivers. Different penalty values are assigned to $\left[0,e_{i}-30\right],\left(e_{i}-30,e_{i}-15\right],\left(e_{i}-15,e_{i}\right],\left(e_{i},l_{i}\right],\left(l_{i},l_{i}+15\right],\left(l_{i}+15,l_{i}+30\right],\left(l_{i}+30,\infty \right]$. In other words, we have two loose time windows $\left[e_{i}-30,l_{i}+30\right]$ and $\left[e_{i}-15,l_{i}+15\right]$ and one tight time window $\left[e_{i},l_{i}\right]$. Discrete penalties can prevent a few patients from suffering from large delays. When $y_{ik}=1$, the penalty is determined using (Equation 3) and (Equation 4); otherwise, $p_{ik}^{a}$ and $p_{ik}^{a}$ are equal to 0. Figure 2 shows the penalties schematically.
(3)$$p_{ik}^{a}=\left\{\begin{array}{c} \beta_{0},a_{ik}\leq e_{i}-30\\ \beta_{1},e_{i}-30<a_{ik}\leq e_{i}-15\\ \beta_{2},e_{i}-15<a_{ik}\leq e_{i}\\ \beta_{3},e_{i}<a_{ik}\leq l_{i}\\ \beta_{4},l_{i}<a_{ik} \end{array}\right.,$$
(4)$$p_{ik}^{d}=\left\{\begin{array}{c} \alpha_{0},d_{ik}\leq l_{i}\\ \alpha_{1},l_{i}<d_{ik}\leq l_{i}+15\\ \alpha_{2},l_{i}+15<d_{ik}\leq l_{i}+30\\ \alpha_{3},l_{i}+30<d_{ik} \end{array}\right.$$

The notation to describe the model is summarized in Table 2.

The objective functions and constraints are formulated by (Equation 5)–(Equation 38).
(5)$$f_{1}=\min\sum_{i,j\in A}\sum_{k\in K}c_{ij}x_{ijk}$$
(6)$$f_{2}=\min\sum_{i\in N}\sum_{k\in K}P_{ik}$$
(7)$$P_{ik}=p_{ik}^{d}+p_{ik}^{a}$$
(8)$$s.t.\sum_{j\in V,i\neq j}\sum_{k\in K}x_{ijk}=1,\forall i\in N$$
(9)$$\sum_{i\in V,i\neq j}x_{ijk}=\sum_{i\in V,i\neq j}x_{jik},\forall k\in K,\forall j\in V$$
(10)$$\sum_{j\in N}x_{0jk}=1,\forall k\in K$$
(11)$$y_{ik}*RC_{i}\leq Q_{k},\forall i\in N,\forall k\in K,\forall Q_{k}\in Q$$
(12)$$y_{ik}=\sum_{j\in V,i\neq j}x_{ijk},\forall i\in N,\forall k\in K$$
(13)$$m\leq \sum_{i\in N}\sum_{j\in V}x_{ijk}\leq n,\forall k\in K$$
(14)$$d_{ik}+t_{ij}\leq a_{jk}+\left(1-x_{ijk}\right)*M,\forall i\in N,\forall j\in V,\forall k\in K,i\neq j$$
(15)$$d_{ik}+t_{ij}\geq a_{jk}-\left(1-x_{ijk}\right)*M,\forall i\in N,\forall j\in V,\forall k\in K,i\neq j$$
(16)$$a_{jk}\leq t_{0j}+\left(1-x_{0jk}\right)*M,\forall j\in N,\forall k\in K$$
(17)$$a_{jk}\geq t_{0j}-\left(1-x_{0jk}\right)*M,\forall j\in N,\forall k\in K$$
(18)$$d_{0k}=0,\forall k\in K$$
(19)$$d_{ik}\geq a_{ik}+\delta_{i},\forall i\in N,\forall k\in K$$
(20)$$d_{ik}\geq e_{i}+\delta_{i},\forall i\in N,\forall k\in K$$
(21)$$d_{ik}\leq a_{ik}+\delta_{i}+\left(1-r_{i}\right)*M,\forall i\in N,\forall k\in K$$
(22)$$d_{ik}\leq e_{i}+\delta_{i}+r_{i}*M,\forall i\in N,\forall k\in K$$
(23)$$\sum_{h\in H}v_{ih}=1,\forall i\in N$$
(24)$$d_{ik}\leq l_{i}*v_{i0}+\left(l_{i}+15\right)*v_{i1}+\left(l_{i}+30\right)*v_{i2}+M*v_{i3},\forall i\in N,\forall k\in K$$
(25)$$d_{ik}\geq l_{i}*v_{i1}+\left(l_{i}+15\right)*v_{i2}+\left(l_{i}+30\right)*v_{i3},\forall i\in N,\forall k\in K$$
(26)$$w_{ik}^{d}=\sum_{h\in H}\alpha_{h}v_{ih},\forall i\in N,\forall k\in K$$
(27)$$p_{ik}^{d}\leq w_{ik}^{d}+M*\left(1-y_{ik}\right),\forall i\in N,\forall k\in K$$
(28)$$p_{ik}^{d}\geq w_{ik}^{d}-M*\left(1-y_{ik}\right),\forall i\in N,\forall k\in K$$
(29)$$p_{ik}^{d}\leq M*y_{ik},\forall i\in N,\forall k\in K$$
(30)$$\sum_{g\in G}u_{ih}=1,\forall i\in N$$
(31)$$a_{ik}\leq \left(e_{i}-30\right)*u_{i0}+\left(e_{i}-15\right)*u_{i1}+e_{i}*u_{i2}+l_{i}*u_{i3}+M*u_{i4},\forall i\in N,\forall k\in K$$
(32)$$a_{ik}\geq \left(e_{i}-30\right)*u_{i1}+\left(e_{i}-15\right)*u_{i2}+e_{i}*u_{i3}+l_{i}*u_{i4},\forall i\in N,\forall k\in K$$
(33)$$w_{ik}^{a}=\sum_{g\in G}\beta_{g}u_{ig},\forall i\in N,\forall k\in K$$
(34)$$p_{ik}^{a}\leq w_{ik}^{a}+M*\left(1-y_{ik}\right),\forall i\in N,\forall k\in K$$
(35)$$p_{ik}^{a}\geq w_{ik}^{a}-M*\left(1-y_{ik}\right),\forall i\in N,\forall k\in K$$
(36)$$p_{ik}^{a}\leq M*y_{ik},\forall i\in N,\forall k\in K$$
(37)$$x_{ijk},y_{ik},r_{i},u_{ig},v_{ih}\in \left\{0,1\right\},\forall i\in V,\forall j\in V,\forall k\in K,\forall g\in G,\forall h\in H,i\neq j$$
(38)$$a_{ik},d_{ik},p_{ik}^{d},p_{ik}^{a},w_{ik}^{d},w_{ik}^{a}\geq 0,\forall i\in N,\forall k\in K$$

The first objective function (Equation 5) is to minimize the travel cost. The second objective function (Equation 6) represents the penalty cost to be minimized. A smaller penalty cost indicates greater satisfaction for both caregivers and patients. Constraints (Equation 8) ensure that a caregiver is assigned to exactly one route. Constraints (Equation 9) mean each caregiver visits the patient and then leaves the patient. Constraints (Equation 10) indicate that caregivers start from the depot and return to the depot after finishing services. Caregivers can perform the service only if their qualification levels are satisfied by constraints (Equation 11) and (Equation 12). Constraints (Equation 13) indicate that each caregiver must serve a certain number of patients in relation to the workload balance. Constraints (Equation 14)–(Equation 17) linearize the Formula (Equation 2). The arrival time at node *j* is the sum of the departure time from node *i* and the travel time from *i* to *j*, when $x_{ijk}=1$. Constraint (Equation 18) specifies that caregivers start their routes from the depot at time 0. The departure times at patients’ locations are defined by constraints (Equation 19)–(Equation 22), which convert the Formula (Equation 1) into linear. Constraints (Equation 14)–(Equation 22) guarantee the schedule feasibility and make subtours impossible. (Equation 23)–(Equation 36) are the variants of (Equation 4) and (Equation 3). Constraints (Equation 37) and (Equation 38) set the domains of decision variables.

For the MILP model, the optimal solutions of each objective function can be obtained by Gurobi Solver. We use a weighted sum method to obtain the approximation of a Pareto optimal set. The sum of the weights of two objectives satisfies $\omega_{1}+\omega_{2}=1$ for normalization. This normalization ensures that the relative importance of each objective is expressed as a fraction of the whole. In the next iteration, $\omega_{1}^{\left(t+1\right)}=\omega_{1}^{\left(t\right)}+\Delta$ and $\omega_{2}^{\left(t+1\right)}=\omega_{2}^{\left(t\right)}-\Delta$. We keep only non-dominated solutions from *T* solutions, which are obtained after *T* iterations.

## 4. Multi-Objective Algorithms

Our proposed method ALNS-EMDLS is divided into two main components: the Enhanced Multi-directional Local Search (EMDLS) described in Section 4.1, and the Adaptive Large Neighborhood Search (ALNS) detailed in Section 4.2. To address uncertain service times, the stochastic version of ALNS-EMDLS is proposed in Section 4.3.

### 4.1. Enhanced Multi-Directional Local Search Algorithm (EMDLS)

The multi-directional local search method was first proposed by Tricoire [51]. Each local search is performed for a single objective (direction) iteratively to improve the non-dominated front *F*. Each local search works separately without considering the importance of the objectives. Only non-dominated solutions are kept after one iteration. This strategy has fewer parameters and can yield well-spread solutions. The savings algorithm is a kind of constructive heuristic and can be used to construct the initial solution. In each direction, we use the ALNS to improve solutions and put the solutions in *F*. The Deb non-dominated sorting method is used to keep the non-dominated front after each iteration. *F* is saved as an ordered list to reduce the number of times executing non-dominated sorting.

Our EMDLS differentiates from the original algorithm in two ways. Firstly, unlike the original algorithm where a single solution from set *F* initiates the next iteration, our approach retains multiple solutions in each direction to enhance diversity. Secondly, our method selects solutions from *F* based on crowding distance, drawing inspiration from NSGA-II [52], rather than making random selections. The crowding distance of point *i* in *F* can be regarded as the perimeter of the hypercube, which is surrounded by the two adjacent points i−1 and i+1. The two boundary points are assigned to a very large number. The solution with a larger crowding distance is more likely to be chosen.

### 4.2. Adaptive Large Neighborhood Search (ALNS)

In the ALNS, various destroy and repair operators are selected adaptively to construct new solutions, which are accepted if their objective function values meet the record-to-record criterion, as detailed in the following subsections.

#### 4.2.1. Destroy and Repair Operators

Three destroy operators and three repair operators are designed based on the previous work in [53]. We remove nodes from the solution by the destroy operators and then insert the removed nodes by the repair operators. To satisfy the constraints (Equation 13), we select the routes with over *m* patients for destruction and those with less than *n* patients for repair. We choose the routes where caregivers meet patients’ demands to respect the constraints (Equation 11).

A certain number of nodes are randomly removed and inserted by the random destroy operator and the random repair operator, respectively. These operators can easily be implemented to run faster than others. The worst destroy operator chooses the nodes with the largest saving that appear to be placed in the wrong position in the solution, while the relatedness destroy operator tends to select the nodes that are similar and can easily be exchanged. The relatedness of the first objective function (Equation 5) can be calculated by $\frac{1}{c_{ij}/c_{\max}+v}$, while the objective function (Equation 6) by $\frac{1}{\left(\right|e_{i}-e_{j}\left|+\right|l_{i}-l_{j}\left|\right)/tw_{\max}+v}$, where $c_{\max}$ denotes the largest cost of all pairs of *i* and *j*, $tw_{\max}$, is the length of the longest time window. If the node *i* and the node *j* are in the same route, v=0; otherwise, v=1. We iteratively find the node with minimum cost position in the greedy repair operator. But for the nodes that are expensive to insert in the last iteration, there are not many opportunities for inserting them because many of the routes are “full”. The regret operator chooses the nodes from the removal set by calculating $i=\arg\max_{i\in u}\left[\sum_{j=1}^{k}\left(\Delta f_{i}^{j}-\Delta f_{i}^{0}\right)\right]$, where *u* is the removal set, and $\Delta f_{i}^{j}$ denotes the insertion value of the node *i* in the $j^{th}$ cheapest insertion position. This method selects the insertion that has a larger possibility to improve the overall performance than the greedy method. Appendix A contains the details of the destroy operators and the repair operators.

#### 4.2.2. Adaptive Weight Adjustment and Acceptance Criterion

Only one destroy operator and one repair operator are chosen by probability $\frac{w_{j}}{\sum_{i=1}^{k}w_{i}}$ in one iteration, where $w_{j}$ is the weight of the $j^{th}$ operator to be chosen, $i\in \left\{1,2,...,k\right\}$. The entire search is divided into several segments. A segment is a number of iterations of the ALNS. The weight $w_{j}$ is automatically updated after a segment and is calculated by the Formula (Equation 39).
(39)$$w_{j}=\left(1-\gamma \right)*w_{i}+\gamma *\frac{r^{\mathrm{score}}}{o^{\mathrm{num}}}$$ The variable $o^{\mathrm{num}}$ means the usage frequency of operator *i* in the latest segment. The reaction factor γ controls how quickly the weight adjustment algorithm reacts to the changes in the effectiveness of the heuristics. The $r^{\mathrm{score}}$ can take three values: $r_{1}$, $r_{2}$, and $r_{3}$, corresponding to three types of acceptance criteria, which assess the heuristic’s recent performance. A high score corresponds to a better performance. More specifically, in the record-to-record method, a neighborhood solution generated by the destroy and repair operators is always accepted if it outperforms the current solution, the best solution, and the sum of the best solution and deviation.

We allow infeasible solutions where some patients in the removal set may not be scheduled by a repair operator. In this case, the number of remaining patients incurs penalties in two objectives by the following formulas:(40)$$\min\sum_{i,j\in V}\sum_{k\in K}c_{ij}x_{ijk}+\eta \sum_{i\in N}z_{i},$$
(41)$$\min\sum_{i\in N}\sum_{k\in K}P_{ik}+\eta \sum_{i\in N}z_{i},$$
where $z_{i}=1$ if the patient *i* can not be inserted; otherwise, $z_{i}=0$.

Algorithm 1 presents each step of the ALNS-EMDLS.
**Algorithm 1** ALNS-EMDLS**Input:** a set *F* only including an initial solution *x*, repair operators, destroy operators, deviation *d*, $iter^{\mathrm{seg}}$, $r_{1}$, $r_{2}$, $r_{3}$**Output:** the Pareto front *F*
1:**repeat**2:   $x^{\mathrm{cur}}\leftarrow crowd\_distance\left(F\right)$3:   $x^{\mathrm{best}}\leftarrow x^{\mathrm{cur}}$4:   G←∅5:   **for** k←1 to *K* **do**6:     **for** j←1 to $iter^{\mathrm{ALNS}}$ **do**7:        **for** i←1 to $iter^{\mathrm{seg}}$ **do**8:          choose $destroy_{i}$ and $repair_{i}$ based on weight *w*9:          $x^{\mathrm{new}}\leftarrow repair_{i}\left(destroy_{i}\left(x^{\mathrm{cur}}\right)\right)$10:          **if** $f_{k}\left(x^{\mathrm{new}}\right)<f_{k}\left(x^{\mathrm{cur}}\right)$ **then**11:             $x^{\mathrm{cur}}\leftarrow x^{\mathrm{new}}$12:             **if** $f_{k}\left(x^{\mathrm{new}}\right)<f_{k}\left(x^{\mathrm{best}}\right)$ **then**13:               $x^{\mathrm{best}}\leftarrow x^{\mathrm{new}}$14:               update score by $r_{1}$15:             **else**16:               update score by $r_{2}$17:             **end if**18:           **else if**
$f_{k}\left(x^{\mathrm{new}}\right)<\left(1+d\right)f_{k}\left(x^{\mathrm{best}}\right)$
**then**19:             $x^{\mathrm{cur}}\leftarrow x^{\mathrm{new}}$20:             update score by $r_{3}$21:          **end if**22:        **end for**23:        update *w* by Formula (Equation 39)24:     **end for**25:     add multiple solutions of optimizing $k^{th}$ objective to G26:   **end for**27:   F←Deb_nondominated_sorting(F,G)28:**until** stopping criterion is met

### 4.3. Stochastic Method

An objective function of a stochastic optimization problem can be written as f(x,Y(ω)), where *x* is the decision variable and *Y* is a random variable that associates a real number to each element ω of a sample space Ω. We simply write it as f(x,ω). Without loss of generality, we consider the form of the stochastic K-objective optimization problem as [54]:(42)$$\min(f_{1}\left(x,\omega \right),f_{2}\left(x,\omega \right),\ldots ,f_{K}\left(x,\omega \right))\;s.t.x\in X.$$ It can be reduced to a deterministic model by defining $f_{i}$ as an *s*-dimensional vector $F_{i}^{\left(s\right)}\left(f_{i}\left(x,\omega \right)\right)$ (for other transformations, see [55,56]). Expectation $E(f_{i}\left(x,\omega \right))$, which is a specific functional $F_{i}^{\left(s\right)}$, is used in this paper. To make the expectation computationally tractable, each objective function is estimated by a sample average approximation method. $E(f_{i}\left(x,\omega \right))$ can be replaced by an unbiased consistent estimator [57].
(43)$$\frac{1}{U}\sum_{\omega_{v}\in S}f_{i}\left(x,\omega_{v}\right),$$
where $S=\{\omega_{1},\omega_{2},\ldots ,\omega_{U}\}$ is a fixed finite set of scenarios drawn in advance.

We assume the service time follows a certain probability distribution that is known in advance. We generate *U* scenarios from the distribution for each patient by means of the Monte Carlo method. The second objective function varies under different scenarios. The first objective function is not calculated by the service time, but it is indirectly affected.

The objective function (Equation 6) can be rewritten as:(44)$${\tilde{f}}_{2}=\frac{1}{U}\sum_{s\in S}\sum_{R_{k}\in R}\sum_{r_{j}\in R_{k}}P_{r_{j}}^{s},$$
where $P_{r_{j}}^{s}$ denotes the penalty of node $r_{j}$ under scenario *s*; $R=\{R_{1},R_{2}\;...,R_{\left|K\right|}\}$ is a set of routes. Here, |K| represents the number of routes, and $R_{k}$ means the $k^{th}$ route in the set *R*. In the stochastic version of ALNS-EMDLS, the second objective function $f_{2}$ is replaced by ${\tilde{f}}_{2}$, and it can be calculated by Algorithm 2 under different scenarios of service time.
**Algorithm 2** Stochastic simulation for computing the expectation of penalty cost**Input:** a solution, a number of scenario *U*, s=1, sum=0**Output:** estimate expected value of penalty cost1:**while** s≤U **do**2:   **for** $R_{k}$ in *R* **do**3:     **for** $r_{j}$ in $R_{k}$ **do**4:        generate ${\tilde{\delta }}_{r_{j}}^{\left(t\right)}$ from sample space according to the probability measure.5:        compute the arrival time and departure time according to (Equation 1) and (Equation 2), then calculate $P_{r_{j}}$ by (Equation 4) and (Equation 3).6:        $sum=sum+P_{r_{j}}^{s}$7:     **end for**8:   **end for**9:**end while**10:the estimated expected value is sum=sum/U

## 5. Computational Study

In this section, our experiment datasets and settings are presented. Some metrics are used to evaluate the quality of non-dominated solutions and the efficiency of multi-objective algorithms. We analyze the results obtained by an exact solution approach (the Gurobi Solver) and two approximate solution approaches (the ALNS-EMDLS and the stochastic ALNS-EMDLS). All of the algorithms have been implemented in Python [58]. For all the experiments, we have used an Intel(R) Core (TM) i5-10310U CPU (@ 2.21 GHz) CPU with 16GB of RAM memory.

The computational study was carried out on two types of data sets. First, and to extend the study of the performance and robustness of the proposed method, a series of tests was created based on literature instances. Second, a several dataset was taken directly from the field. It was used to analyze the proposed solutions in terms of business practices and to draw out managerial insights.

### 5.1. Data Sets and Experimental Setup

No benchmark results exist in the literature for our problem. Hence, we generate six different types of (C1, C2, R1, R2, RC1, and RC2) instances based on the Solomon dataset [59], which contains various instances with different sizes and characteristics for the VRPTW problem. This variety allows for comprehensive testing of our algorithm across different instances. The Solomon dataset provides necessary details such as locations, time windows, and service times, aligning well with our proposed problem. The data sets that support the findings of this study are openly available in “figshare” at http://doi.org/10.6084/m9.figshare.21339072 (accessed on 1 March 2024). We have compared problems involving 25, 50, and 100 patients, focusing on the characteristics including the geographical data, the length of the scheduling horizon, and the proportion of time-constrained patients. The geographical data are randomly generated in R1 and R2, clustered in C1 and C2, and a mix of both structures in RC1 and RC2. The sets R1, C1 and RC1 have a short scheduling horizon, while the sets R2, C2, and RC2 have a longer one. Each type comprises four sets. For example, C1 consists of C1-a, C1-b, C1-c, and C1-d. The only distinction between C1-a and the other three sets lies in the presence of some patients having time windows that are scarcely constrained in sets C1-b, C1-c, and C1-d.

Each patient is available only between the ready time and the due time. Distance is Euclidean, and the value of the travel time is equal to the value of distance between two nodes. We assign random values to patient requirements for caregiver levels and service times. To be more practical, we diversity the service time of each patient. The mean value of service time δ accounts for twenty to sixty percent of the time window. We assume that the service time of each patient is an independently normally distributed random variable and follows $N(\mu_{i},\sigma_{i}^{2})$. We set μ=δ and $\sigma =\sqrt{\frac{\delta }{5}}$. We assume that three caregivers are assigned to 25 patients, five caregivers to 50 patients, and 10 caregivers to 100 patients.

Before starting the problem-solving, the parameters are set. For solving the MILP by Gurobi Solver, we set Δ as 1/50. The ALNS is affected by random factors, so we utilize the average value of the metrics from five runs for every experiment. In each iteration, the number of nodes removed by destroy operators is randomly set between 2 and 4, as reflected in the variable *q* in Algorithms A1 and A2 (refer to Appendix A). η in Formula (Equation 40) is set to 1000. Given the impracticability of testing all hyperparameter combinations, we employ Bayesian optimization for efficient exploration [60]. It is assumed the hyperparameters are in a black box (an unknown function), with the function’s output evaluated via a metric known as the hypervolume indicator (detailed in Section 5.2). In Bayesian optimization, it assumes the unknown function stems from a Gaussian process prior, updating the posterior distribution with new observations. An acquisition function is chosen for the next evaluation point. The tuned hyperparameters of the proposed method and their best values after 50 iterations are shown in Table 3.

### 5.2. Performance Metrics

The metrics are used to measure the convergence and diversity (diversity includes ductility and uniformity) of the solutions [61], including the number of Pareto optimal points (N), the hypervolume indicator (HV), and the Spread metric (S) [62]. These metrics provide a comprehensive assessment of an algorithm’s performance in identifying a diverse and well-distributed set of solutions.

The objective values are scaled between 0 and 1 before calculating these metrics. The HV gives the volume *x* enclosed by a reference point and the solutions and is shown in the following Formula (Equation 45):(45)$$HV={⋃}_{x\in A}V(x,R).$$ The reference point *R* is commonly set to the point (1,1) for minimizing a bi-objective problem. This metric measures convergence and diversity. A larger value of hypervolume signifies the better quality of the solutions. The spread metric evaluates the diversity of solutions and is given by:(46)$$sp=\frac{d_{f}+d_{l}+\sum_{i}^{n-1}|d_{i}-\bar{d}|}{d_{f}+d_{l}+\bar{d}\left(n-1\right)},$$
where $d_{f}$ and $d_{l}$ are the Euclidean distances between the extreme solutions in true Pareto front and non-dominated solutions (NDS). $\bar{d}$ is the average of the whole distance $d_{i}$, and $d_{i}$ is the Euclidean distance between one solution and the next nearest solution, where i∈[1,|NDS|−1].

We assume the extreme values of a true Pareto front are (0,1) and (1,0). Smaller values indicate better distribution.

### 5.3. Deterministic Bi-Objective Solutions

This is a base case in which the service times are considered as deterministic quantities. We compare the ALNS-EMDLS and the Gurobi Solver to measure the performance of our proposed method.

The two extreme points of Pareto front ([$f_{1}^{\min}$, $f_{2}$] and [$f_{2}^{\min}$, $f_{1}$]), HV, S, the CPU execution time of the program measured by seconds (TCPU), and the number of Pareto points (N) for the small size (10 patients) and the real-life size (25, 50, and 100 patients) instances are shown in Table 4. We find that the running time of the Gurobi Solver is much longer than that of the ALNS-EMDLS. The extreme values of the two objectives are very close. The outcomes produced by the proposed method, which encompass a broad-range Pareto front, are remarkably comparable to those of the Gurobi Solver while requiring less time. Moreover, the results of the ALNS-EMDLS involve more solutions than the Gurobi Solver. Real-world instances typically contain a larger number of patients. With large instances, solving the problem to optimality is troublesome, and we found no solution within a limited time using the Gurobi Solver. We, therefore, considered the ALNS-EMDLS. Computation times shown in Table 4 indicate that the ALNS-EMDLS is effective. Other metrics show the Pareto fronts achieved by the proposed method are well distributed and have satisfactory diversity.

The numerical simulation processes of stochastic programming require a lot of time. Utilizing the Gurobi Solver to derive solutions for the stochastic model would require significantly more time compared to the deterministic model. The proposed method is adequate to find satisfactory solutions. The performances of the proposed method are very close to the best-found solutions obtained by the Gurobi Solver. Therefore, for large instances, we propose employing the stochastic version of ALNS-EMDLS to solve the problem under uncertainty.

### 5.4. Stochastic Bi-Objective Solutions

Table 5 shows the metrics and extreme objective function values of the Pareto front of the Stochastic ALNS-EMDLS (S_EMDLS).

On average, a considerable difference exists between the solution for minimum travel cost and the solution for minimum penalty cost in terms of both objectives. Hence, the quality of services that decision-makers decide to offer to patients has a significant impact on operating costs, underscoring the need for careful decision-making. The instances C1 and C2, which are clustered, have lower minimum travel costs than R and RC. We compared the results under different lengths of time windows. Data types 2, including C2, R2, and RC2, have longer time windows than type 1. Table 5 shows that most of the values of the minimum penalty of type 2 are less than those of type 1.

The data sets 25-C1-a, 25-C1-b, 25-C1-c, and 25-C1-d are used to test the effect caused by different percentages of patients with time windows. In 25-C1-a, all patients are available only at certain periods of the day. 28%, 52%, and 68% patients are available for the full working time of caregivers for 25-C1-b, 25-C1-c, and 25-C1-d, respectively. Most patients do not have time windows in the 25-C1-d set. The Pareto fronts are shown in Figure 3. The solutions of 25-C1-d dominate 25-C1-a, 25-C1-b, and 25-C1-c.

We use Algorithm 2 to evaluate the solutions obtained by the Deterministic (original) ALNS-EMDLS (D*_EMDLS). The number of solutions and HV and S metrics are compared in Table 6.

It can be inferred from the results that the modeling of uncertain service times will increase computing time because the S_EMDLS needs to handle more information than only one scenario. Using the proposed stochastic framework, more scenarios contribute to the output. The S_EMDLS is therefore more realistic. In Table 6, the results of the ANOVA test show there is no significant difference in the means of HV and S between the solutions of D*_EMDLS and S_EMDLS since the *p*-values are greater than 0.005. This means that the stochastic method can yield solutions that are at least as good as the deterministic one. The proposed stochastic approach is designed to optimize the expected values of objective functions. Its solutions may not be global optimal solutions for the individual scenario, but they are robust, providing possible realizations despite uncertain service times.

## 6. Management Recommendations

### 6.1. The Influence of Uncertainty on Cost and Care Quality

To identify the behavior of the proposed model and method and examine the influence of uncertain service times on objective values, several sensitivity analysis are performed on the main parameters. In this regard, a small test problem of 25 patients and three caregivers is selected. The parameters include the ending time of the loose time windows (ET) and variance of distribution (VD), which can indicate the range of uncertain service times. Each parameter has three levels, namely, small, medium, and large. To validate the robustness of solutions, we also compare the results of D*_EMDLS and S_EMDLS. We use the solutions obtained by the deterministic model to evaluate their sensibilities under uncertain service times. We normalize the Pareto set to [0,1] and calculate the distance between the origin and each point in the Pareto set. The solution with a minimum distance (D) is defined as a trade-off solution in our case. The objective values (travel cost (TC), penalty (P), normalized travel cost (NTC), and normalized penalty (NP)) of the trade-off solutions and minimum distances are summarized in Table 7 and Table 8.

$l_{i}$ means the latest time of the tight time window. If the departure time somehow exceeds the time window by a certain level, there will be a penalty cost. If the departure time lies within ($e_{i}$, $l_{i}$], ($l_{i}$, $l_{i}+c_{1}$], ($l_{i}+c_{1}$, $l_{i}+c_{2}$] or ($l_{i}+c_{2}$, *∞*] ($c_{1}<c_{2}$), the penalty cost will be 0, $\alpha_{0}$, $\alpha_{1}$, $\alpha_{2}$, $\alpha_{3}$, respectively (see Figure 2 and Formula Equation 4). That is to say, we have loose time windows. If $c_{1}$ and $c_{2}$ are bigger, patients give more flexibility to the decision-makers and caregivers. Three levels of ET are compared when the VD is δ∗2 in Table 7 and Table 8. The results of the two methods do not dominate each other when the ET levels are small and medium. But the result of D*_EMDLS is dominated by S_EMDLS when $c_{1}=30$ and $c_{2}=45$. If patients have more flexibility, the S_EMDLS is better to deal with uncertain service times.

Regarding VD, sensitivity analysis has been performed by increasing the variance of normal distribution. We sample the service times from normal distributions. If the variance is bigger, which means patients are more likely to have larger or smaller service times that deviate from the average value, the results of S_EMDLS dominate D*_EMDLS (shown in the last rows of Table 7 and Table 8). Figure 4 shows the Pareto points of D*_EMDLS and S_EMDLS with different VD and ET. The short lines in the box plots denote the median of TC or P. In (e), when VD=δ∗2, $LD=(l_{i}+30,l_{i}+45)$, the median of S_EMDLS is obviously smaller than D*_EMDLS. However, the S_EMDLS is more realistic as it takes into account multiple scenarios, leading to increased computing time. If the variance of service times is not too large, the D_EMDLS can be chosen to save computing time.

The values of travel cost and penalty of trade-off solutions for different approaches are shown in Table 9 and Figure 5. The values of travel cost of the D*_EMDLS are less stable than those of S_EMDLS. When VD increases from δ/3 to δ∗2.5, the penalty of S_EMDLS increases by 28.24%, while the growth for the D*_EMDLS is 43.32%. The penalty of S_EMDLS changes more sluggishly than that of D*_EMDLS when VD changes. Therefore, S_EMDLS has better robustness than the D*_EMDLS. The objective values are more stable and perform better for the cases with large variations in service times by using the S_EMDLS, while D_EMDLS is more appropriate for cases with small variations in service times to save computing time. Decision-makers can select one of the solutions from the Pareto sets depending on their companies’ operating profitability. They can select which method to use in order to attain better objective values based on varying conditions.

### 6.2. Practical Application for Enhanced Understanding

In this section, we apply the results derived from Section 6.1 to a real-life case and provide some actionable management recommendations for choosing schedules.

Using the real-life data provided by “Soins et Santé”, a home health care company located in Lyon, France, we implement our methods to create routes and schedules. A total of 27 patients receive home care services. The available data set includes patients’ locations, service times, and available time sessions. The duration of each service varies, ranging from 5 to 46 minutes. The patients are visited in three time sessions: morning sessions from 7:30 to 12:00, afternoon sessions from 13:00 to 15:30, and evening sessions from 17:00 to 19:30. We create the time windows based on the patient’s preferred time sessions for visits. The length of the time windows ranges from 30 to 150 minutes. We also create the required level of caregiver for each patient.

Figure 6 shows the results when VD=δ∗2, $LD=(l_{i}+30,l_{i}+45)$, m=4, and n=10. In this figure, the Pareto front of the S_EMDLS dominates that of the D*_EMDLS algorithm. The S_EMDLS is capable of generating a more diverse range of solutions. In real-life cases, when implementing the solution obtained by the D_EMDLS, if the difference between the real service times and the planned service times is small, the actual objective function values are close to the original ones. However, if the real service times are significantly different from the planned service times, the solution obtained by the S_EMDLS can be chosen to achieve smaller objective values on average. If it is not possible to determine the extent to which the patients’ service times differ from the planned service times, the solution obtained by S_EMDLS can be used by decision-makers to have higher stability.

By providing a more comprehensive view of the various solutions within the Pareto front, managers can gain a deeper understanding of the strengths and weaknesses of the solutions in the Pareto front. This knowledge can then be used to make decisions on selecting the most appropriate solution for their needs. We examine three points on the Pareto front, namely, the solution with the minimum travel cost ($S_{1}$), the trade-off solution ($S_{2}$), and the solution with the minimum penalty ($S_{3}$). These points are clearly marked with three star icons within Figure 6. We chose the trade-off solution with objective values closest to the origin. Table 10 presents the objective values and their respective indicators. The indicator $PER_{e}$ represents the percentage of patients who are visited by caregivers before the earliest time of their time windows. The indicator $PER_{l}$ denotes the percentage of patients whose services are completed by caregivers after the latest time of their time windows. The indicators $WT_{min}$ and $WT_{max}$ mean the minimum and maximum daily working hours, respectively. The values of $f_{1}$ for $S_{1}$ and $S_{2}$ are lower compared to $S_{3}$, whereas $f_{2}$ and $PER_{l}$ for $S_{1}$ and $S_{2}$ exceed those of $S_{3}$. This reveals that $S_{3}$ offers improved service punctuality but leads to higher travel costs. The values of $WT_{min}$ and $WT_{max}$ for $S_{3}$ are lower than those for $S_{1}$ and $S_{2}$, signifying shorter waiting times for caregivers, contributing to a more equitable distribution of workload. The routes corresponding to these solutions are visualized in Figure 7. The locations of the patients (expressed by their longitude and latitude) and the planned routes are displayed on a map with a blank background to ensure their anonymity. The routes of $S_{3}$ are too complicated to be constructed manually, and our model and method proposed a useful tool to construct them. We offer actionable recommendations for choosing solutions as follows.

If the majority of patients have a higher tolerance for exceeding their end time of the time windows, the manager may opt for a solution located on the left side of the Pareto front, which prioritizes minimizing travel costs.Although the minimum and maximum numbers of patients that each caregiver needs to visit (*m* and *n*) are limited in the proposed model, the solution $S_{3}$ can be chosen to achieve a better workload balance.The routes of $S_{3}$ can be selected when the manager prefers better satisfaction for patients and caregivers.

## 7. Conclusions

We developed a bi-objective model for the HHC problem. This model aims to optimize the travel costs and satisfaction of both patients and caregivers, considering several practical constraints: the soft time windows, the matching of patient needs and caregiver skills, and the workload balance. Additionally, we consider uncertain service times to enhance the practicality and robustness.

To solve the bi-objective optimization problem, we developed an ALNS-EMDLS to obtain Pareto fronts. The Stochastic ALNS-EMDLS (S_EMDLS) was proposed to deal with the problem under the uncertain service times. First, we considered only one deterministic scenario: the average value of service times sampling from the normal distribution. The comparison between the Gurobi Solver and the ALNS-EMDLS revealed the latter’s superior efficiency and competitively high-quality solutions. Second, we considered uncertain service times, assuming they follow the normal distribution. In the D*_EMDLS, the solutions obtained by the ALNS-EMDLS were evaluated under uncertain service times. The results showed that when the two parameters, i.e., the ending time of the loose time and variance of distribution, are bigger, the S_EMDLS outperforms since its solutions dominate those of D*_EMDLS. We evaluated the trade-off solutions of D*_EMDLS and S_EMDLS under varying variances. The outcomes confirmed S_EMDLS’s robustness, effectively demonstrating its efficacy in managing uncertain service times. Finally, a real-life application was conducted to provide practical managerial suggestions for choosing routes and schedules.

As a future development, we will create new heuristics and use exact methods to compare the results of this study. More practical objectives and constraints motivated by the needs of HHC companies can be added to further studies. We will try to accelerate the computing time as the S_EMDLS takes over twenty times longer to implement than the D_EMDLS. At present, the study is conducted on a daily planning horizon; in the future, mid-term or long-term planning of HHC activities will be considered. For future applications, we aim to ensure seamless model integration into the current HHC management system, focusing on real-time data for more accurately and effectively predicting future uncertainties.

## Figures and Tables

**Figure 1 ijerph-21-00377-f001:**
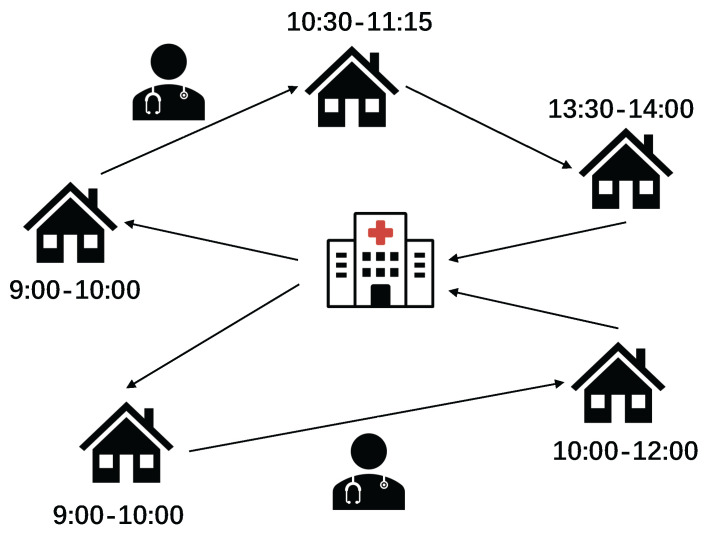
Daily activities in home health care.

**Figure 2 ijerph-21-00377-f002:**
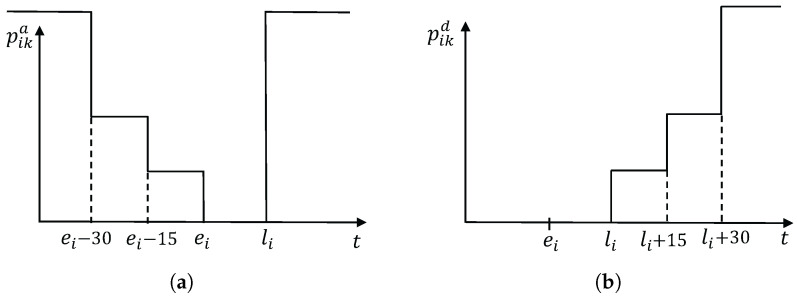
Illustration of discrete penalty. (**a**) Discrete penalty for arrival time; (**b**) discrete penalty for departure time.

**Figure 3 ijerph-21-00377-f003:**
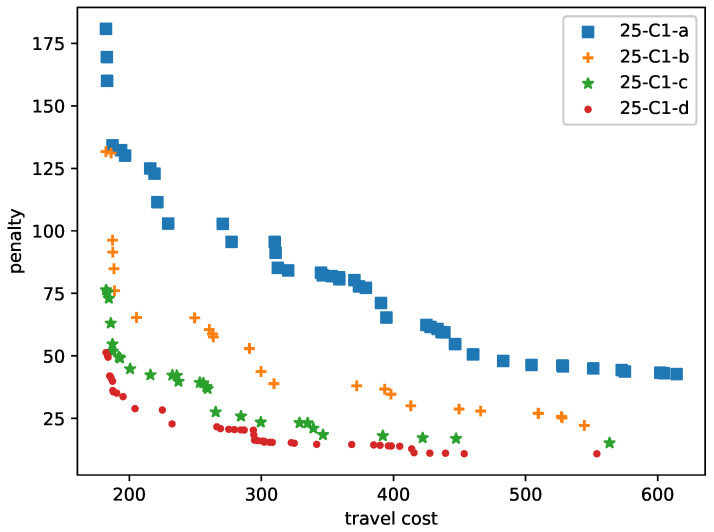
Time windows comparison of 25 patients.

**Figure 4 ijerph-21-00377-f004:**
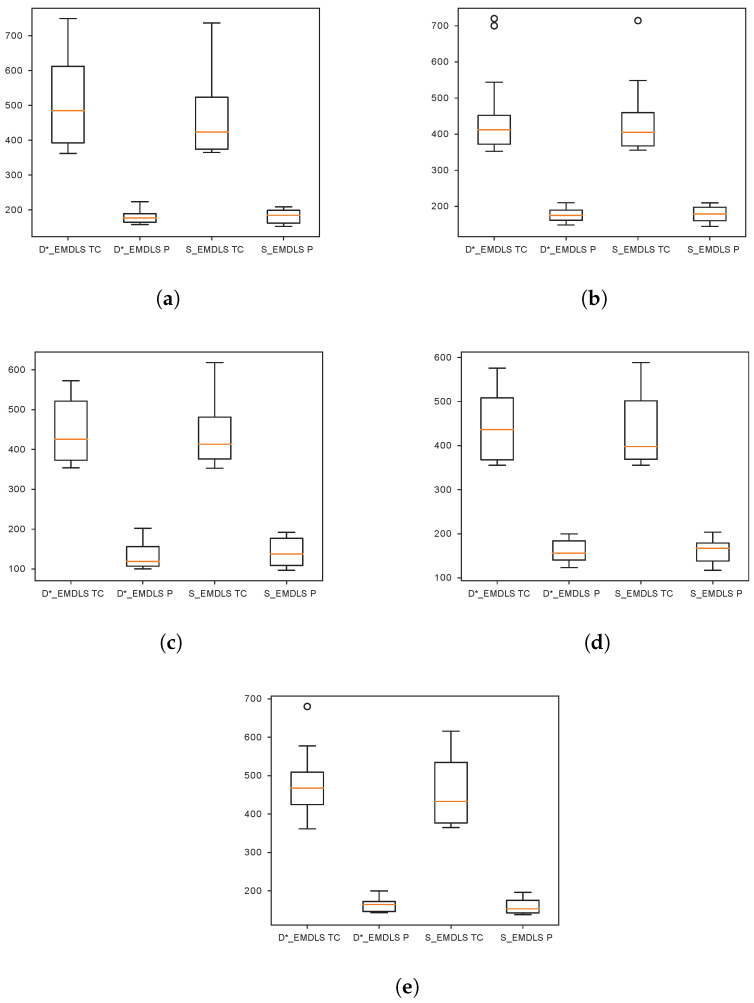
TC and P with different VD and ET. (**a**) $VD=\delta *2,LD=(l_{i}+5,l_{i}+15)$; (**b**) $VD=\delta *2,LD=(l_{i}+15,l_{i}+30)$; (**c**) $VD=\delta /3,LD=(l_{i}+30,l_{i}+45)$; (**d**) $VD=\delta ,LD=(l_{i}+30,l_{i}+45)$; and (**e**) $VD=\delta *2,LD=(l_{i}+30,l_{i}+45)$.

**Figure 5 ijerph-21-00377-f005:**
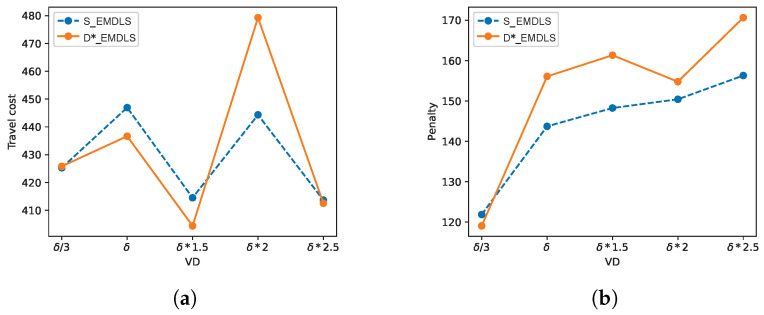
Objective values of trade-off solutions with changing of VD. (**a**) Travel cost; (**b**) penalty.

**Figure 6 ijerph-21-00377-f006:**
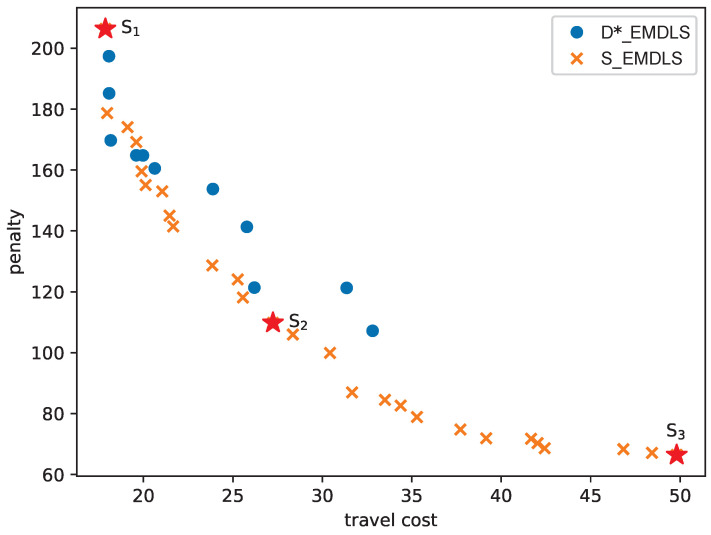
Pareto front on a real-life case.

**Figure 7 ijerph-21-00377-f007:**
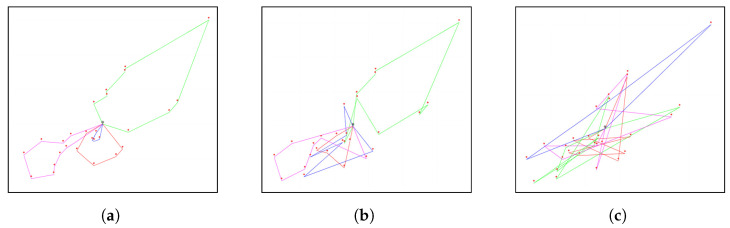
Routes displayed on a blank background map. (**a**) $S_{1}$; (**b**) $S_{2}$; and (**c**) $S_{3}$.

**Table 1 ijerph-21-00377-t001:** Deterministic models and methods used in the latest research.

Author	MOO	TW	WB	QC	Other	Methods
Liu, Yuan, and Jiang [12]		×			Lunch break	B&P
Shahnejat-Bushehri et al. [13]		×		×	Idle time, synchronization	SA, TS
Trautsamwieser et al. [14]		×		×	Over time, break time	VNS
Bertels and Fahle [15]		×	×	×		CP, TS
Decerle et al. [28]	×	×	×		Synchronization	MAMO
Braekers et al. [31]	×	×		×	Patients inconvenience	MDLS
Our study	×	×	×	×		ALNS-EMDLS

MOO, Multi-Objective Optimization; TW, Time Windows; WB, Workload Balance; QC, Quality of Caregivers; B&P, Branch and Price; SA, Simulated Annealing; TS, Tabu Search; VNS, Variable Neighborhood Search; CP, Constraint Programming; MAMO, Memetic Algorithm for Multi-objective Optimization; and MDLS, Multi-Directional Local Search.

**Table 2 ijerph-21-00377-t002:** Notation.

Notation	Definition
Sets	
N	set of patients
V	set of depot and patients
A	all arcs
K	set of caregivers
Q	set of levels of qualification
RC	set of requirements of patients for levels of caregivers
H	set of number of intervals divided by departure time
G	set of number of intervals divided by arrival time
Parameters	
*i*,*j*	index of patients
*k*	index of caregivers
$c_{ij}$	travel cost between *i* and *j*
$t_{ij}$	travel time between *i* and *j*, is $c_{ij}$
$Q_{k}$	level of qualification of caregiver *k*
$RC_{i}$	requirement of patient *i* for qualification level of a caregiver
$\delta_{i}$	service time of patient *i*
$e_{i},l_{i}$	time window of patient *i*
*m*,*n*	minimal number of patients and maximal number of patients that one caregiver is able to visit
$\alpha_{h}$	degree coefficient if departure time is located at $h^{th}$ interval
$\beta_{g}$	degree coefficient if arrival time is located at $g^{th}$ interval
Decision variables	
$x_{ijk}$	binary decision variable: 1 if caregiver *k* moves from *i* to *j*, 0 otherwise
$y_{ik}$	binary decision variable: 1 if patient *i* is served by caregiver *k*, 0 otherwise
$a_{ik}$	arrival time of caregiver *k*’s visit to patient *i*
$d_{ik}$	departure time that caregiver *k* leaves patient *i*
$p_{ik}^{d},p_{ik}^{a}$	continuous decision variable: the penalties that arrival time and departure time are outside of time windows
$w_{ik}^{d},w_{ik}^{a}$	auxiliary variables: continuous, $p_{ik}=w_{ik}*y_{ik},\forall i\in N,\forall k\in K$
$u_{ig},v_{ih}$	binary decision variable: 1 if caregivers’ arrival (departure) time at patient *i* is located at $g^{th}$ ($h^{th}$) interval, 0 otherwise
$r_{i}$	binary decision variable: 1 if caregiver arrives after $e_{i}$

**Table 3 ijerph-21-00377-t003:** Hyper parameters.

Notation	Definition	Value
$r_{1}$	score if $f\left(x^{\mathrm{new}}\right)<f\left(x^{\mathrm{cur}}\right)$	21.38
$r_{2}$	score if $f\left(x^{\mathrm{new}}\right)<f\left(x^{\mathrm{best}}\right)$	18.93
$r_{3}$	score if $f\left(x^{\mathrm{new}}\right)<\left(1+d\right)f\left(x^{\mathrm{best}}\right)$	7.08
γ	coefficient of weight (see Formula (Equation 39))	0.68
$iter^{\mathrm{seg}}$	the number of iterations to update weight	4
$iter^{\mathrm{ALNS}}$	the number of iterations of ANLS of each direction	19
*d*	percentage of the objective value of the best solution	0.13

**Table 4 ijerph-21-00377-t004:** Results of Gurobi Solver and ALNS-EMDLS.

	min $f_{1}$		min $f_{2}$					
	$f_{1}^{\min}$	$f_{2}$	$f_{2}^{\min}$	$f_{1}$	**HV**	**S**	**N**	**TCPU**
Gurobi Solver								
10-C1	128.64	52.00	32.00	147.69	0.72	0.15	4.00	293.13
10-C2	163.35	62.00	39.00	189.59	0.61	0.09	4.00	625.47
10-R1	194.47	85.00	21.00	258.70	0.68	0.07	6.00	233.86
10-R2	194.47	68.00	18.00	344.84	0.66	0.06	7.00	271.53
10-RC1	218.06	81.00	23.00	336.60	0.74	0.12	8.00	285.31
10-RC2	230.23	75.00	18.00	462.66	0.68	0.02	10.00	293.19
ALNS-EMDLS								
10-C1	128.64	52.00	33.00	153.64	0.74	0.15	4.00	21.35
10-C2	163.35	62.00	39.00	189.59	0.62	0.07	6.00	20.55
10-R1	194.47	85.00	21.00	258.70	0.69	0.12	8.00	23.15
10-R2	194.47	68.00	18.00	344.84	0.68	0.00	18.00	26.70
10-RC1	218.06	81.00	23.00	336.60	0.75	0.04	16.00	31.90
10-RC2	230.23	75.00	18.00	462.66	0.71	0.004	15.00	26.11
%	**0.00 **	**0.00**	0.005	0.007	2.45	−29.02	65.08	−91.60
ALNS-EMDLS								
25-C1	182.33	108.00	12.50	527.05	0.76	0.06	25.75	64.08
25-C2	240.45	131.75	26.00	497.09	0.86	0.04	19.00	59.19
25-R1	352.94	162.00	48.75	500.59	0.66	0.05	19.00	60.40
25-R2	350.88	166.50	7.00	820.81	0.80	0.03	24.25	63.59
25-RC1	294.99	135.50	5.25	386.91	0.71	0.04	26.75	58.32
25-RC2	294.99	149.00	3.75	935.51	0.78	0.04	28.25	61.94
50-C1	344.90	226.25	20.50	1220.02	0.72	0.02	39.50	138.76
50-C2	447.31	176.00	27.50	1476.68	0.72	0.02	34.00	134.35
50-R1	564.50	360.25	139.25	947.44	0.77	0.06	29.50	136.68
50-R2	570.02	251.00	2.50	1419.08	0.73	0.02	36.00	139.71
50-RC1	529.73	235.00	26.00	665.48	0.63	0.04	29.50	144.86
50-RC2	591.95	253.75	1.75	1721.49	0.73	0.03	37.50	145.97
100-C1	823.09	414.25	38.75	3297.75	0.68	0.01	56.25	256.97
100-C2	887.19	501.25	41.75	3417.91	0.74	0.01	54.00	241.71
100-R1	935.09	621.75	141.00	1526.37	0.70	0.01	44.25	249.72
100-R2	941.73	530.50	14.25	2692.53	0.72	0.02	55.25	256.78
100-RC1	1006.43	613.50	108.25	1610.18	0.70	0.02	42.00	249.80
100-RC2	1019.04	468.50	8.50	3326.02	0.74	0.03	48.75	256.62

**Table 5 ijerph-21-00377-t005:** Results of S_EMDLS.

	min $f_{1}$		min $f_{2}$					
**S_EMDLS**	$f_{1}^{\min}$	$f_{2}$	$f_{2}^{\min}$	$f_{1}$	**HV**	**S**	**N**	**TCPU**
25-C1	182.35	110.05	22.73	569.10	0.82	0.06	37.25	1318.99
25-C2	239.94	131.46	28.49	597.37	0.85	0.02	27.25	1256.51
25-R1	349.69	179.25	49.63	516.97	0.75	0.02	20.75	1176.40
25-R2	352.78	121.94	13.03	923.21	0.76	0.02	39.75	1306.94
25-RC1	294.99	114.43	9.89	390.71	0.67	0.01	37.25	1196.68
25-RC2	294.99	140.60	6.24	1129.41	0.82	0.02	41.25	1357.98
50-C1	347.27	219.11	39.74	1375.92	0.80	0.02	47.50	2985.87
50-C2	449.06	199.25	33.66	1618.00	0.78	0.02	47.50	2766.20
50-R1	568.73	298.00	143.21	1020.79	0.81	0.05	26.00	3034.78
50-R2	566.30	248.64	15.29	1624.68	0.75	0.02	49.00	2954.02
50-RC1	539.62	222.38	40.75	737.78	0.74	0.04	31.50	2819.57
50-RC2	570.14	231.47	5.62	2235.57	0.76	0.02	48.25	2991.65
100-C1	831.97	485.78	73.61	3591.33	0.74	0.01	63.75	6370.87
100-C2	916.35	459.38	65.52	3754.04	0.77	0.01	61.75	5635.48
100-R1	948.95	599.82	157.89	1758.63	0.69	0.01	40.25	5790.88
100-R2	947.37	539.87	42.95	2942.96	0.75	0.01	64.00	6022.64
100-RC1	1013.65	559.13	122.02	1727.37	0.70	0.02	45.00	5824.40
100-RC2	1016.62	463.38	19.54	3973.91	0.75	0.01	68.00	6000.49

**Table 6 ijerph-21-00377-t006:** Deterministic model tested on uncertain environment.

	min $f_{1}$		min $f_{2}$		
**D*_EMDLS**	$f_{1}^{\min}$	$f_{2}$	$f_{2}^{\min}$	$f_{1}$	**HV**
25-C1	182.33	107.10	26.62	476.12	0.77
25-C2	240.45	126.21	30.70	542.94	0.87
25-R1	352.94	142.54	52.70	480.16	0.61
25-R2	350.88	150.13	16.83	826.24	0.79
25-RC1	294.99	122.17	10.97	386.25	0.68
25-RC2	294.99	139.35	10.41	947.57	0.78
50-C1	344.90	223.08	50.22	1141.53	0.74
50-C2	447.32	175.01	41.00	1452.68	0.74
50-R1	564.50	329.10	151.81	961.09	0.80
50-R2	570.02	232.62	25.47	1423.64	0.74
50-RC1	529.73	220.87	49.27	653.99	0.62
50-RC2	591.95	244.33	12.24	1814.66	0.74
100-C1	823.09	412.46	96.72	3250.29	0.71
100-C2	887.20	493.36	73.05	3258.56	0.74
100-R1	935.09	579.99	173.21	1544.58	0.68
100-R2	941.73	501.50	65.53	2739.26	0.73
100-RC1	1006.43	567.45	144.81	1623.38	0.68
100-RC2	1019.04	458.41	35.52	3588.20	0.76
ANOVA					
F	0.00	0.01	0.37	0.26	2.51
p	0.98	0.92	0.55	0.61	0.12

**Table 7 ijerph-21-00377-t007:** Results of different levels of ET and VD using D*_MDLS.

D*_MDLS	Levels	D	NTC	NP	TC	P
ET						
Small	$l_{i}+5$, $l_{i}+15$	0.38	0.33	0.18	491.23	169.40
Medium	$l_{i}+15$, $l_{i}+30$	0.33	0.28	0.18	453.71	159.82
Large	$l_{i}+30$, $l_{i}+45$	0.40	0.37	0.16	**479.28**	**154.78**
VD						
Small	δ/3	0.38	0.33	0.18	425.81	119.06
Medium	δ	0.57	0.37	0.43	436.65	156.08
Large	δ∗2	0.40	0.37	0.16	**479.28**	l**154.78**

**Table 8 ijerph-21-00377-t008:** Results of different levels of ET and VD using S_MDLS.

S_EMDLS	Levels	D	NTC	NP	TC	P
ET						
Small	$l_{i}+5$, $l_{i}+15$	0.39	0.34	0.19	492.77	163.08
Medium	$l_{i}+15$, $l_{i}+30$	0.34	0.30	0.15	464.14	154.40
Large	$l_{i}+30$, $l_{i}+45$	0.37	0.32	0.20	**444.33**	**150.42**
VD						
Small	δ/3	0.38	0.27	0.26	425.29	121.88
Medium	δ	0.50	0.39	0.31	446.94	143.72
Large	δ∗2	0.37	0.32	0.20	**444.33**	**150.42**

**Table 9 ijerph-21-00377-t009:** Objective values of trade-off solutions with changing of VD.

Methods	δ/3	δ	δ∗1.5	δ∗2	δ∗2.5
Travel cost					
D*_EMDLS	425.81	436.65	404.40	479.28	412.53
S_EMDLS	425.29	446.94	414.51	444.33	413.67
Penalty					
D*_EMDLS	119.06	156.08	161.34	154.78	170.64
S_EMDLS	121.88	143.72	148.26	150.42	156.30

**Table 10 ijerph-21-00377-t010:** Indicators for three solutions.

Solutions	$f_{1}$	$f_{2}$	${\mathrm{PER}}_{e}$	${\mathrm{PER}}_{l}$	${\mathrm{WT}}_{\min}$	${\mathrm{WT}}_{\max}$
$S_{1}$	17.87	206.36	28.96%	71.11%	298.99	806.27
$S_{2}$	27.24	109.82	52.30%	35.19%	424.29	714.94
$S_{3}$	49.81	66.42	45.33%	28.30%	263.57	677.99

## Data Availability

The data sets that support the findings of this study are openly available in “figshare” at http://doi.org/10.6084/m9.figshare.21339072 (accessed on 1 March 2024).

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
