# Peer review of "A Bi-Objective Home Health Care Routing and Scheduling Problem under Uncertainty"

_ijerph, 2024, doi:10.3390/ijerph21030377_

Round 1

Reviewer 1 Report

Comments and Suggestions for Authors

Comments on the Quality of English Language

no comments.

Author Response

Reviewer #1:

  1. Please complete the sentence in the line 159.

Answer: Thank you for your useful suggestion. We added a sentence to link the context. (See page 4 line 162)

“Some multi-objective optimization algorithms are capable of obtaining Pareto solutions, which represent the set of non-dominated solutions in a multi-objective optimization problem. These solutions are essential for decision-makers to evaluate trade-offs between conflicting objectives.”

  1. Please explain this:

Answer: It denotes the service time of patient . We have added the explanation on page 7 line 295 and Table 2.

“The caregiver will leave immediately after serving for service time δi.”

  1. Give citation for Gurobi Solver.

Answer: Done. (See page 6 line 267)

  1. Explain why . A beginner scholar should understand this scientific concept.

Answer: By setting the sum of weights equal to 1, we normalize the weights. This ensures that the relative importance of each objective is expressed as a fraction of the whole. We added this explanation on page 10 line 360.

“The sum of the weights of two objectives satisfies ω1 + ω2 = 1 for normalization. This normalization ensures that the relative importance of each objective is expressed as a fraction of the whole.”

  1. In line 286, A penalty cost is introduced in objective 286 functions when caregivers arrive or leave outside the time window. A valid reason should specify about it.

Answer: The penalty cost is defined as the objective function (6). If the arrival times are close to the earliest time of the time window and the departure times are close to the latest time of the time window, caregivers do not need to wait too long while patients receive timely care. Minimizing the penalty cost helps ensure service punctuality and helps improve the satisfaction of caregivers and patients. We also added a sentence to clarify the importance of penalty cost. (See page 7 line 300)

“Minimizing the penalty can enhance service punctuality and reduce waiting times, thereby improving the satisfaction of both patients and caregivers.”

  1. All constraints from (14)-(22) are to be explained as authors explained constraints from (1)-(13).

Answer: We reorganized and clarified the explanation of the constraints. The constraints are the linearization of (1) and (2). (See page 7 line 296 and page 10 line 350)

  1. Which language you have used to perform numerical experiments. Please specify about it with proper citation.

Answer: We have used Python. We added one sentence on page 13 line 460.

  1. In Algorithm 2, what is U?

Answer: U is the number of scenarios which has been explained on page 13 line 447. Now it's also specified in the algorithm 2 as an input.

  1. Explain the datasets and their attributes? Why did authors choose these datasets?

Answer: The Solomon dataset is famous for the VRPTW problem and contains various instances with different numbers of customers and different geographical data. Our proposed problem is an extension of the VRPTW problem. It includes the main information we need for our proposed problem, and its diversity allows for comprehensive testing of our algorithm across different instances. We added some sentences on page 14 line 471.

“Hence, we generate six different types of (C1, C2, R1, R2, RC1, RC2) instances based on the Solomon dataset [59], which contains various instances with different sizes and characteristics for the VRPTW problem. This variety allows for comprehensive testing of our algorithm across different instances. The Solomon dataset provides necessary details such as locations, time windows, and service times, aligning well with our proposed problem.”

Reviewer 2 Report

Comments and Suggestions for Authors

Home health care companies provide health care services to patients in their homes. Due 1 to increasing demand, the provision of home health care services requires effective management 2 of operational costs while satisfying both patients and caregivers. In practice, uncertain service 3 times might lead to considerable delays that adversely affect service quality. To this end, this paper proposes a new bi-objective optimization problem to model the routing and scheduling problems 5 under uncertainty in home health care, considering the qualification and workload of caregivers. 6 A mixed-integer linear programming formulation is developed. Authors should follow the given comments:

1.       Please complete the sentence in the line 159.

2.       Please explain this:

3.       Give citation for Gurobi Solver.

4.       Explain why . A beginner scholar should understand this  scientific concept.

5.        In line 286, A penalty cost is introduced in objective 286 functions when caregivers arrive or leave outside the time window. A valid reason should specify about it.

6.       All constraints from (14)-(22) are to be explained as authors explained constraints from (1)-(13).

7.       Which language you have used to perform numerical experiments. Please specify about it with proper citation.

8.       In Algorithm 2, what is ?

9.       Explain the datasets and their attributes? Why did authors choose these datasets?

I must state that the manuscript can be accepted after giving the proper response to the above comments. 

Author Response

Reviewer #2:

  1. On Page 3. line 121: "Simulated Annealing (SA) and Tabu Search (TS) were applied in two phrases." - The term "phrases" seems to be a typo. It's likely intended to be "phases." referring to the stages or steps in which these algorithms were applied.

Answer: Thank you for your careful check. We replaced it with "phases".

  1. On Page 4, line 135: The concept of a "segmented penalty" for arrival and departure times outside of time windows is introduced but not fully explained. Providing a brief example or a more detailed description of how penalties are segmented could help readers understand the potential benefits of this approach, such as more evenly distributed penalties and increased fairness for patients and caregivers.

Answer: We added a sentence on page 4 line 138 according to your comments. It is also explained on page 7 line 305.

“If the penalty is continuous, only one patient is likely to endure a long delay. The segmented penalty promotes a more equitable distribution of service punctuality and increases the fairness in scheduling for patients and caregivers.”

  1. On Page 6, line 221: the term "uncertain theory" might be confusing or appear as a typo. If it was intended to be "uncertainty theory"?

Answer: Thank you for your careful check. We replaced it with " uncertainty theory ".

  1. On Page 6, lines 246-250: the paper does a great job of situating its contributions within the landscape of existing research. However, specifying how the scenario-based stochastic method differs in application from the single-objective optimization problems previously focused on by other studies could further highlight the novelty of the current research.

Answer: Thanks for your useful suggestions. We added one sentence to highlight the benefits of the scenario-based stochastic method on page 6 line 258. The advantage of the multi-objective model has been explained in line 260.

“The scenario-based stochastic method is data-driven and enhances decision-making under uncertainty by evaluating multiple scenarios, allowing for more informed strategies [47]. The multi-objective optimization model and algorithm, that we proposed, offer a broader range of choices where manager preferences are unknown.”

  1. Section Problem statement, each constraint is described briefly in relation to its purpose within the model. Expanding on the rationale behind key constraints, especially those unique to the HHCRSP (like qualification matching and time window adherence), could provide readers with a better understanding of their significance and how they contribute to addressing the problem.

Answer: Qualification matching ensures service safety. Certain health care tasks must be performed by individuals with specific qualifications and caregivers are matched with tasks that fit their skill sets. Patients often require services within specific time frames, which could be due to medical needs, personal preferences, or both. The time window constraint ensures that services are scheduled within or close to these specified intervals. It helps improve the satisfaction of both caregivers and patients. We added more explanation on page 7 line 286.

“The skill level of nurses is mainly based on their experience in dealing with complex cases. Patients are assured of receiving appropriate levels of care. The qualification matching ensures the service quality and safety.”

  1. The model incorporates real-world complexities such as caregiver qualifications and patient time windows. Further discussing how these elements are derived from real operational scenarios in home health care could strengthen the connection between the mathematical model and its practical application.
  2. Answer: The skill level of nurses is mainly based on their experience in dealing with complex cases. Different levels of caregivers are needed according to the patient’s health condition to ensure service quality. Patients specify their preferred time frame for home care during the registration process. But care services like insulin injections, require a fixed time frame that is determined by the physician. We added some sentences on page 7 line 285 and line 290.

“A visit is therefore allowed only if the patient’s requirement is lower or equal to the qualification level of the caregiver.”

“Patients indicate their preferred time for home care during the registration process.”

  1. The metrics used to evaluate the solutions, including the number of Pareto optimal points, Hypervolume indicator, and Spread metric, are appropriately chosen for a multi-objective optimization problem. It might be helpful to briefly explain why these specific metrics were chosen over others, especially for readers who may not be familiar with multi-objective optimization literature.

Answer: These metrics can be used to evaluate the convergence and diversity of the algorithm.

on page 15 line 510.  The feature of each metric has been detailed on page 15 line 513-525.

“These metrics provide a comprehensive assessment of an algorithm’s performance in identifying a diverse and well-distributed set of solutions.”

  1. The section 6.2 offers actionable recommendations but could benefit from a clearer outline of step- by-step actions that managers can take based on the model's findings. Providing examples of specific decisions made using the model's output could illustrate its utility further.

Answer: We added some analysis of Table 10, reorganized the sentences, and listed the actionable recommendations (see page 21 line 661).  Figure 7 displays the model's outputs, from which the manager can select one of three proposed solutions. These solutions outline the routes that caregivers can follow to visit patients.

“The values of f1 for S1 and S2 are lower compared to S3, whereas f2 and PERl for S1 and S2 exceed those of S3. It reveals that the S3 offers improved service punctuality but leads to higher travel costs. The values of WTmin and WTmax for S3 is smaller than those for S1 and S2, signifying shorter waiting times for caregivers, contributing to a more equitable distribution of workload. The routes corresponding to these solutions are visualized in Fig. 7. The locations of the patients (expressed by their longitude and latitude) and the planned routes are displayed on a map with a blank background to ensure their anonymity. The routes of S3 are too complicated to be constructed manually, and our model and method proposed a useful tool to construct them. We offer actionable recommendations for choosing solutions as follows.

  1. If the majority of patients have a higher tolerance for exceeding their end time of the time windows, the manager may opt for a solution located on the left side of the Pareto front, which prioritizes minimizing travel costs.
  2. Although the minimum and maximum numbers of patients that each caregiver needs to visit (m and n) are limited in the proposed model, the solution S3 can be chosen to achieve a better workload balance.
  3. The routes of S3 can be selected when the manager prefers better satisfaction for patients and caregivers.”

  1. While the section 6.2 focuses on a specific case study, discussing the generalizability of the recommendations to other settings or contexts within the home health care industry would be valuable. This could include considerations for different sizes of operations, geographic locations, or service types.

Answer: Thank you for your useful suggestions. Our results can also be generalized to applications in waste collection, where drivers visit each customer’s residence or the disposal site. These points have different available time windows, and the model considers minimizing the travel time of the drivers while also taking into account the acceptance hours. However, the main focus of this paper is on home health care. We choose to mention the various applications of the VRPTW in the literature review. (See page 4 line 105)

“The VRPTW has many applications such as telecommunication, waste collection and cross-docking [8 ,9 ].”

  1. A brief discussion on any limitations observed during the application of the model to the real-life scenario and the challenges faced could offer a more comprehensive view. It would also be helpful to suggest how these limitations might be addressed in future work.

Answer: Integration to the management system and real-time data acquisition can be focused in the future. (see page 23 line 713)

“For future applications, we aim to ensure seamless model integration into the current HHC management system, focusing on real-time data for more accurately and effectively predicting future uncertainties.”